# Critical porosity of gas enclosure in polar firn independent of climate

Christoph Florian Schaller[1], Johannes Freitag[1], and Olaf Eisen[1,2]

[1]Alfred Wegener Institute, Helmholtz Centre for Polar and Marine Research, D-27568 Bremerhaven, Germany
[2]Department of Geosciences, University of Bremen, D-28359 Bremen, Germany

*Correspondence to:* Christoph Florian Schaller (christoph.schaller@awi.de)

**Abstract.** In order to interpret the paleoclimatic record stored in the air enclosed in polar ice cores, it is crucial to understand the fundamental lock-in process. Within the porous firn, bubbles are sealed continuously until the respective horizontal layer reaches a critical porosity. Present-day firn air models use a postulated temperature dependence of this value as the only parameter to adjust to the surrounding conditions of individual sites. However, no direct measurements of the firn microstructure could confirm these assumptions. Here we show that the critical porosity is a climate-independent constant by providing an extensive data set of µm-resolution 3D X-ray computer tomographic measurements for ice cores representing different extremes of the temperature and accumulation ranges. We demonstrate why indirect measurements suggest a climatic dependence and substantiate our observations by applying percolation theory as a theoretical framework for bubble trapping. Incorporation of our results does significantly influence the dating of trace gas records, changing gas age–ice age differences by up to more than 1,000 years. This may further help resolve inconsistencies, such as differences between East Antarctic $\delta^{15}$N records (as a proxy for firn height) and model results. We expect our findings to be the basis for improved firn air and densification models, leading to lower dating uncertainties. The reduced coupling of proxies and surrounding conditions may allow for more sophisticated reinterpretations of trace gas records in terms of paleoclimatic changes and will benefit the development of new proxies, such as the air content as a marker of local insolation.

## 1 Introduction

Air trapped in polar ice cores provides a unique opportunity for paleoclimatic studies (Legrand and Mayewski, 1997). In particular, it allows reconstruction of the past chemical and isotopic composition of the atmosphere for up to 800,000 years (Jouzel et al., 2007; Loulergue et al., 2008). However, as bubbles are only isolated from the atmosphere in a certain depth, the firn-ice transition (50–120 m depending on the local conditions), the enclosed air is always younger than the surrounding ice. Accurate estimation of this gas age–ice age difference ($\Delta$age), up to 7,000 years during glacial periods (Bender et al., 2006), is essential for the interpretation of ice-core records as otherwise phase relationships between ice and gas records cannot be determined correctly.

Thus, it is crucial to understand the fundamental processes in the porous firn (Schwander and Stauffer, 1984) – diffusion of air through the open pore space (Trudinger et al., 1997; Fabre et al., 2000) and the entrapment of air by pore closure due to firn

densification, which is the main focus of this study. In a depth range referred to as the lock-in zone, gas enclosure within individual horizontal layers occurs at a critical porosity (Schwander et al., 1993). It is the only parameter in empirical relations of closed and total porosity (Schwander, 1989; Goujon et al., 2003) that are commonly used in present-day firn air models (Severinghaus and Battle, 2006; Mitchell et al., 2015). A temperature dependence of this value has been postulated (Raynaud and Lebel, 1979) and parametrized using air-content measurements (Martinerie et al., 1992). Nonetheless, the underlying microstructural processes are not well-understood and there is no confirmation of these assumptions by direct measurements of firn microstructure.

The $\delta^{15}N$ of $N_2$ has been established as a proxy for firn height and thus an indirect constraint on $\Delta$age (Sowers et al., 1992). This relation has successfully been tested for high-accumulation sites, e.g. the last 40,000 years at Summit, Greenland (Schwander et al., 1997). On the contrary, there is a mismatch of up to 2,000 years with model results for the East Antarctic plateau (Bender et al., 2006; Parrenin et al., 2012). These modeled chronologies are based on the current knowledge of bubble trapping in polar firn and particularly sensitive to the critical porosity via the assumed temperature dependence. Deviations from the simple relationships used to reconstruct past temperatures and accumulation rates from the water isotopic composition have been suggested as a possible explanation (Landais et al., 2006), while the hypothesis of a large glacial convective zone as an important factor has been ruled out (Capron et al., 2013). Recently the inclusion of impurity effects has reduced the mismatch for East Antarctic sites, however it deteriorates the agreement between modelled and measured $\delta^{15}N$ for high-accumulation sites (Breant et al., 2017).

In this paper, we present the first extensive data set of direct firn microstructure measurements throughout the lock-in zone. We start off by using it to scrutinize the current knowledge of gas enclosure in polar firn and show why previous indirect measurements yielded different results. Then, we apply bond percolation theory (Enting, 1993) as a theoretical framework for our conclusions and demonstrate their agreement with other methods. Finally, we discuss changes in the dating and interpretation of trace-gas records that incorporation of our results in current firn air models will imply. The reduced coupling of proxies and surrounding conditions may allow for more sophisticated reinterpretations in terms of paleoclimatic changes and will benefit the development of new proxies, such as the air content as a marker of local insolation (Raynaud et al., 2007; Eicher et al., 2016).

## 2 Materials and Methods

Firn microstructure throughout the lock-in zone has been deduced for ice cores from three locations (cf. Table 1) using a specifically designed X-ray microfocus computer tomograph in a cold lab (Freitag et al., 2013). For each one meter core segment, we scanned a minimum number of five sections of approximately 4 cm height and the full core diameter (8–10 cm) with a focus on homogenous layers. One measurement consists of 3,000 radioscopic images, which are used to tomographically reconstruct the 3D microstructure at a resolution of approximately 25 μm (e.g. Fig. 1b). Consecutively, these reconstructions are segmented into ice and air using a two-step procedure consisting of a two-level Otsu's method (Otsu, 1979) followed by simple region growing for the ambiguous voxels. We adapted an existing algorithm (Nguyen et al., 2011) to determine

the pore coordination number during the segmentation process. To eliminate the effect of cut pores at the surface of the sample (Martinerie et al., 1990), each data point (as referred to e.g. in Table 1) corresponds to a layer of approximately 1 cm height and 6 cm diameter. Having the microstructure of the surrounding material in all directions at hand allows us to safely determine whether a pore is open or closed. For all measurements, the remaining cut pores were less than $0.1\%$ of the pore
volume. For ten repeat measurements of the same sample, both standard and maximum deviation of the total porosity are less than $1\%$. Furthermore, the total porosities agree with those from bulk measurements and 2D radioscopy with a maximum deviation of $3\%$.

A well-known framework to model porous media is bond percolation theory (Broadbent and Hammersley, 1957). It enables us to predict the point at which a material becomes impermeable. The Kelvin structure (packed tetrakaidecahedra, see Fig. 1a)
is space-filling with one of the lowest surface-area-to-volume ratios. It is well-studied and has for example been applied as a model for foam (Koehler et al., 1999). We use it to represent sintered ice grains. When packed, the grains align along a body-centered cubic lattice. Therefore the air network corresponds to its dual lattice which has a coordination number (average number of neighbors) of four when fully occupied. For this lattice, the fraction of channels occupied by air at gas enclosure, the so-called percolation threshold, is known to be $0.4031$ (van der Marck, 1997). Thus the predicted coordination number at
the percolation threshold is $4 \cdot 0.4031 = 1.6124$. Notably, the influence of the chosen lattice is rather small (Wierman and Naor, 2003).

## 3   Results

Gas enclosure within a single layer occurs at the same critical porosity $\Phi_{\text{crit}}$ of about $0.1$ for all cores (Fig. 2a). However, as indicated by the much steeper slope of the closed porosity, enclosure takes place in a significantly smaller porosity range for
the East Antarctic cores compared to the coastal Greenland site. To fit our data, we derived a new local relation (Eq. (1)) of closed porosity $\Phi_{\text{cl}}$ and total porosity $\Phi$, where $b, \lambda_1, \lambda_2 \in \mathbb{R}_{\geq 0}$ and $b \leq 1$. The parameters of least squares fitting are given in Table 2.

$$\Phi_{\text{cl}} = \begin{cases} \Phi & \text{for } \Phi \leq \Phi_{\text{crit}} \\ \Phi_{\text{crit}} \left( b\, e^{-\lambda_1 (\Phi - \Phi_{\text{crit}})} + (1-b)\, e^{-\lambda_2 (\Phi - \Phi_{\text{crit}})} \right) \text{ else} \end{cases} \tag{1}$$

Within the microstructure analysis for the B53 core, we also mimicked the sample properties (cylindrical shape, 5 cm diameter,
5 cm height) and the method (melting the sample under vacuum conditions, thus counting cut closed pores as part of the open pore space) as applied for Summit, Greenland (Schwander et al., 1993). This significantly changes the shape of the closed versus total porosity curve and yields results similar to previous studies (Fig. 2b). Then, by comparing with our original data (where cut pores are traced within a larger volume to determine whether they are open or closed), we determined the necessary correction factors for the effect of cut pores (Fig. 4).

For the coordination number (Fig. 1c) we observe a linear increase with total porosity for all three sites. At the critical porosity of about $0.1$ we obtain very similar values of $1.65 \pm 0.17$ for B53, $1.7 \pm 0.18$ for B49 and $1.64 \pm 0.24$ for RECAP_S2 from linear regression.

## 4 Discussion

Even though the surrounding conditions differ significantly, we obtain the same critical porosity of about $0.1$ for all cores (Fig. 2a, Table 2). In previous literature, average ice densities at air isolation were obtained from air-content measurements on deep ice samples (Martinerie et al., 1992). To allow for a better comparison with our results, we calculated the corresponding critical porosities (Fig. 3). For the gas enclosure within single layers, we do not observe the commonly assumed temperature dependence of $\Phi_{\mathrm{crit}}$. In contrast, we find strong evidence for a constant (and thus climate-independent) critical porosity.

In order to estimate the closed porosity in firn, previous studies relied on measuring the amount of air enclosed in a sample by melting it in a vacuum chamber. However, during vacuumization, air is not only removed from the open pore space, but also cut closed pores, which is of particular importance for the more extensive pore network of the firn compared to deeper ice samples. Breaking of closed, but still fragile pores might even enhance this effect (Schwander and Stauffer, 1984). Nonetheless it has been neglected or only accounted for by multiplying with correction factors of up to $10\%$ to date (first applied for firn

in Appendix 2 of Martinerie et al. (1992); recently Mitchell et al. (2015)). Our estimation (Fig. 2b and Fig. 4) proves a serious underestimation of the cut-pore effect. This can be explained by a classical percolation phenomenon – near the percolation threshold, individual (clusters of) closed pores can be very large compared to single bubbles (Stauffer, 1979). Indeed, we observe extents of more than a centimeter near the critical porosity for all three cores.

In particular, our results confirm the existence of a critical porosity in contrast to recent assumptions of gas enclosure for a

single layer occurring within a certain porosity range (Mitchell et al., 2015). Remarkably, for the correct critical porosity, the Schwander parametrization (Schwander, 1989) seems to approximately represent a site-independent average relation of closed and total porosity (cf. Fig. 2a). However, due to the lack of other parameters, it cannot fully reflect the behavior of polar firn. Therefore we decided to derive a more complex exponential-decay relation (Eq. (1)) to fit our results.

For all three cores, the observed coordination numbers at gas enclosure (Fig. 1c) are in agreement with the value predicted

by percolation theory. We conclude that polar firn evolves towards the same "optimal" microstructure, driven by a universal percolation process (Enting, 1993). However, the initial conditions differ as the firn is strongly influenced by the surrounding local conditions such as accumulation rate (affecting residence times in certain depth intervals) and temperature (as one of the main drivers for snow and firn metamorphism (Schneebeli and Sokratov, 2004)).

The problem of understanding gas enclosure in polar firn has been tackled with different methods and from various perspec-

tives such as firn microstructure, firn air transport and firn air pumpings. As a consequence, seemingly different definitions have been established for terminological frameworks such as the "lock-in zone". The results of two firn air pumpings conducted at the RECAP drill site (T. Sowers, personal communication, 2017) and at Kohnen station, close to B49 (Weiler, 2008) in combination with high-resolution X-ray porosity measurements (Freitag et al., 2013), corroborate our microstructural findings. For

both sites, the sharp decline in $CO_2$, $CH_4$ and $N_2O$ concentrations (interpreted as the onset of the lock-in zone according to firn air pumpings) coincides with the occurrence of the first significant (i.e. at least 1 cm thick) layer with a porosity below the critical value of $0.1$. On the other hand, no more air can be pumped (bottom of the lock-in zone according to firn air pumpings) when there are no further layers with a total porosity larger than $0.1$. In the firn air transport literature (e.g. Buizert and

Severinghaus, 2016), the onset of the lock-in zone (also refered to as "lock-in depth") is defined as the depth where molecular diffusion effectively ceases. According to percolation theory this happens at the percolation threshold, i.e. the point when there is no longer a connected component of the order of the system size (Ghanbarian and Hunt, 2014). This corresponds to the first layer reaching a closed porosity of $100\%$, which is the onset of the lock-in zone in the microstructural sense. Regarding the bottom of the lock-in zone, it has been observed that due to vertical mixing the air composition in a certain depth does only no

longer change (definition according to gas transport) at the "close-off depth" (Buizert et al., 2012). It is defined as the depth at which all pores are closed (Witrant et al., 2012) and thereby also coincides with the bottom of the lock-in zone according to firn microstructure. Thus, the three definitions for the lock-in zone (according to firn microstructure, firn air transport and firn air pumpings) are equivalent. Furthermore, the limits of the lock-in zone are solely determined by the existence of significant layers above and below the critical porosity, and thereby the (cm-scale) porosity variability.

While concepts such as gas enclosure (both in single layers and as a bulk property) occurring at a critical closed (Goujon et al., 2003) or open (Gregory et al., 2014) porosity have become widely accepted, they do not seem to agree with the results of our firn microstructure analysis and the previously discussed (conceptual) definitions of the lock-in zone. As a consequence, refinement of these theories may greatly benefit the understanding of gas enclosure in polar firn. Notably, the critical closed porosity value of $37\%$ identified by Jean-Marc Barnola using porosity measurements of several ice cores from Greenland and

Antarctica (Goujon et al., 2003) corresponds to a total porosity of approximately $0.1$ for the two data sets (Summit and B53) that are affected by the cut-pore effect (displayed in Fig. 2b).

Even though gas enclosure for a single layer occurs at the same critical porosity, sealed layers may have variable air contents. Above the close-off depth, we determine average coefficients of variation for the total porosity of $1.3\%$ for B53, $1.8\%$ for B49 and $2.5\%$ for RECAP_S2. Higher porosity variability will lead to a larger amount of shallowly trapped pores, thereby increasing

the air content $V$ (Stauffer et al., 1985). In our case, the effect of shallow trapping can be estimated from the different slopes of the lock-in curves given in Fig. 2a, yielding possible increases in air content of about $2\%$ for B49 and $8\%$ for RECAP_S2 in comparison with B53. This implicitly assumes that closed and open porosity undergo the same compaction as the firn densifies and thus has to be interpreted as the maximum possible influence of shallow trapping. In addition, the lock-in zone extends over a depth range of approximately 7 m for B53, 9 m for B49 and 15 m for RECAP_S2. Larger lock-in zones are expected

to cause enhanced sealing effects (i.e. permeable layers being sealed by impermeable ones above). This further increases the air content (Stauffer et al., 1985). However, the effect is hard to quantify as our measurements do not yield information about the spatial extent of horizontal layers and it does not take into account pressure adjustment within the lock-in zone which is happening on a much shorter time scale compared to diffusion (Buizert and Severinghaus, 2016). Nonetheless, it may explain the $8\%$ and $27\%$ larger air contents for B49 and RECAP_S2 (compared to B53) respectively, that $V$ measurements for deep ice

cores would predict according to the observed temperature dependence (Martinerie et al., 1992). In return, even though we do

not observe this temperature dependence for the gas enclosure within single layers, it is a signal that seems to originate from the lock-in zone, presumably as a consequence of a distinct density layering.

We conclude that $V$ measurements may yield multiple-layer averages of pore volumes at gas enclosure. They should only be interpreted with great caution in regards to the sealing of single layers. The post-coring loss of enclosed air is an error source we can neither quantify nor rule out. For the Camp Century core about $10\%$ lower air contents were observed after 35 years of storage (Vinther et al., 2009), although a systematic error due to the different measurement setups is possible.

## 5 Implications

For the EDC core (East Antarctica), $86\%$ of the variance in $V$ cannot be explained by air pressure or temperature changes. An anti-correlation with local insolation was found and suggested as a new proxy (Raynaud et al., 2007). The same insolation signature was found for the $V$ record of the NGRIP core (Greenland), but the underlying physical mechanisms are not yet resolved (Eicher et al., 2016). Based on our results, we rule out the idea of other properties influencing the porosity at gas enclosure for single layers as we do not even observe a temperature dependence. Instead, we suggest increased sealing effects and shallow trapping due to larger porosity variability of the layered snowpack as an explanation. Reasons for the enhanced layering may be changes in the atmospheric conditions, accumulation rate or impurity content, similar to the observed increase in layering during glacials (Augustin et al., 2004).

As indicated by $\delta^{15}$N measurements as a proxy for firn height (Sowers et al., 1992), up-to-date firn air models seem to have difficulties to estimate past lock-in depths for the East Antarctic plateau (Landais et al., 2006; Capron et al., 2013) and to synchronize age dating of individual ice cores (Parrenin et al., 2012). We suggest that incorporation of our results will help to overcome these problems, as current approaches are based on temperature-dependent lock-in (Martinerie et al., 1992) and the Barnola model (Goujon et al., 2003). Exemplarily, we estimate the gas age–ice age difference for the Vostok ice core from the temperature (Jouzel et al., 1987) and accumulation rate (Parrenin, 2004) records using the Herron-Langway model (Herron and Langway, 1980). On average, excluding the temperature dependence of the critical porosity reduces the gas age–ice age difference by well over $10\%$. For the last glacial more than 1,000 years of the 2,000 year mismatch with $\delta^{15}$N data (Bender et al., 2006) can be explained this way. We suggest a combination with the effect of impurities on firn densification (Freitag et al., 2013; Breant et al., 2017) as a promising approach to resolve the remaining mismatch. Other effects that are currently not well represented, such as stronger layering during the glacials (Bendel et al., 2013), may further influence these values. We see this study as a catalyst for improved firn air and densification models, that will reduce dating uncertainties and allow for more sophisticated reinterpretations of the available trace gas records, in particular due to the reduced coupling to temperature.

*Code and data availability.* The data shown in the plots are available through the open-access library PANGAEA$^{\circledR}$. If you are interested in using our implementation of the described algorithms or want to work with the raw data, please contact the main author.

*Author contributions.* JF was responsible for the development of the AWI ICE-CT and pointed out the opportunity for this study to CS. 3D measurements were carried out by JF for RECAP_S2 and CS for B49 and B53. The segmentation of the 3D data sets and the evaluation of microstructural parameters was performed by CS, who researched and programmed the necessary algorithms. The results and their implications were discussed and related to the literature by all coauthors. CS prepared the initial manuscript, which was reviewed and improved by all coauthors.

*Competing interests.* The authors declare no conflict of interest.

*Acknowledgements.* The authors want to acknowledge Sepp Kipfstuhl and Bo Vinther as the responsibles for the drilling of B49/B53 and RECAP_S2 respectively and Todd Sowers for providing an insight into the RECAP firn-air data. The main author wants to thank the German National Merit Foundation (Studienstiftung des deutschen Volkes e.V.) for funding his PhD project. Last but not least, we are much obliged to Eric Wolff, Christo Buizert and an anonymous referee for their support in greatly improving this manuscript.

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

**Table 1.** Details on the analyzed cores. Mean annual temperature is denoted by $\overline{T}$ and an estimate of yearly accumulation by $\dot{a}$.

| Drill site | Year | Elevation [m] | $\overline{T}$ [°C] | $\dot{a}$ [kg m$^{-2}$ a$^{-1}$] | Depth interval [m] | No. of data points |
|---|---|---|---|---|---|---|
| RECAP_S2 (Renland, Greenland)[†] | 2015 | 2296 | -18 | 460 | 49–73 | 246 |
| B49 (Kohnen station, East Antarctica)* | 2012/13 | 2881 | -44 | 65 | 73–90 | 303 |
| B53 (Dome Fuji, East Antarctica)* | 2012/13 | 3726 | -55 | 30 | 76–106 | 614 |

[†] Johnsen et al. (1992)

* Unpublished data, the accumulation rates are based on a preliminary volcanic layer dating.

**Table 2.** Results of least squares fitting our parametrization to the obtained data.

| Data set | $\Phi_{crit}$ | $\lambda_1$ | $\lambda_2$ | $b$ | $R^2$ |
|---|---|---|---|---|---|
| RECAP_S2 | 0.1005 | 62.45 | 47.34 | 0.4816 | 0.9744 |
| B49 | 0.0985 | 169.57 | 51.55 | 0.5797 | 0.9801 |
| B53 | 0. 1000 | 206.36 | 48.06 | 0.7072 | 0.9603 |

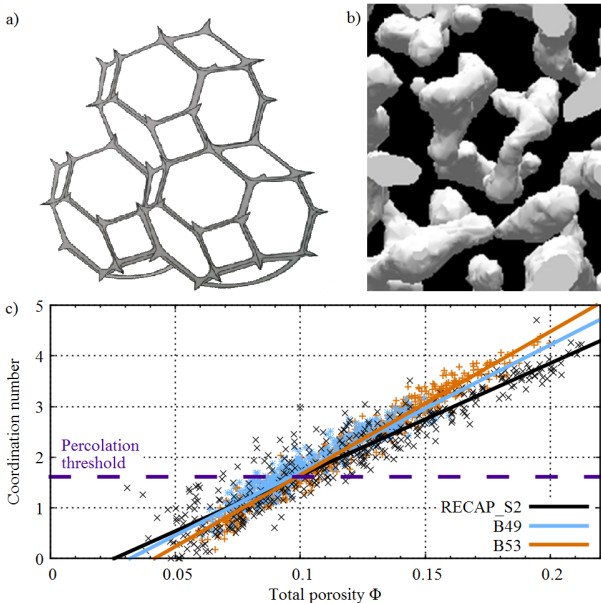

**Figure 1.** a) A structure consisting of three packed tetrakaidecahedra. The white bodies do represent ice crystals, the gray edges the pore network. b) Example of a 3D scan, the pore network is shown in white. c) Coordination number versus total porosity for our measurements. The threshold for gas enclosure within a single layer as predicted by percolation theory has been marked.

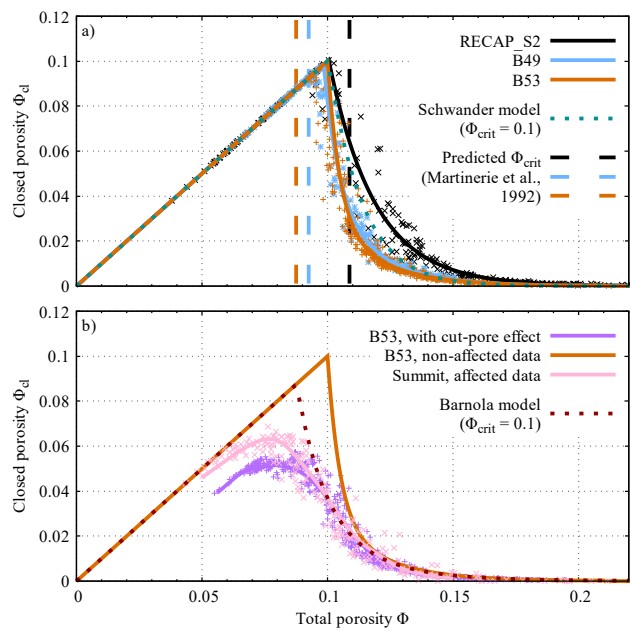

**Figure 2.** Closed versus total porosity for a) the analyzed cores in comparison with commonly expected values and b) B53 ignoring cut pores compared to previous results from Summit, Greenland (Schwander et al., 1993). The solid lines indicate least squares fits for the respective core, the short-dashed lines represent model results (Schwander, 1989; Goujon et al., 2003) for the given parameters and the long-dashed lines mark the critical porosity values predicted by the previously observed temperature dependence (Martinerie et al., 1992).

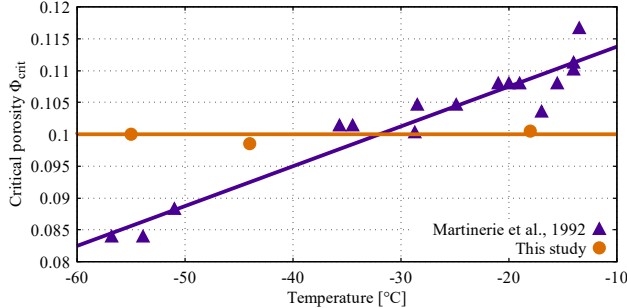

**Figure 3.** Critical porosity versus temperature. The given linear relation is commonly fit to the data of Martinerie et al. (1992), a study based on air-content measurements of 495 deep ice samples from sixteen cores (with a minimum of only two measurements for one core). From their results, they reconstruct the average ice density at air isolation which is equivalent to the porosities shown here. In contrast, we analyzed the microstructure of 1163 firn samples for three cores (see Table 1), allowing the direct determination of the critical porosity of gas enclosure within a single layer.

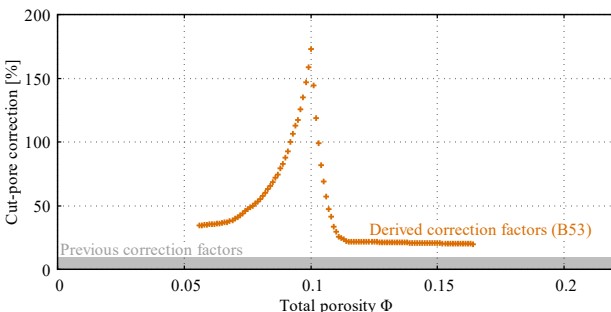

**Figure 4.** Necessary correction factors for the cut-pore effect assuming the B53 core would have been analyzed with the method used for Summit, Greenland (Schwander et al., 1993). The range of corrections applied in previous literature is indicated by the gray box.