# Peer review of "Critical porosity of gas enclosure in polar firn independent of climate"

_Climate of the Past, 2017_

## Short Comment (SC1) · 10 Aug 2017

It is interesting to see further work on applying the percolation model from lattice statistics for the analysis of bubble trapping in firn.

This paper probably needs to take more account of a defining characteristic of such transitions: the existence of fluctuations on all length scales. It was the observation of large fluctuations in closed porosity by Schwander and Stauffer (1984) that suggested to Enting (1985) that percolation may be a useful description of firn closure. The simulations given by Enting (1993) show, for various numbers of bubble sites, the degree of sample to sample variation that occurs. The value of the Schaller et al paper would be greatly enhanced if the results reported the number of pore sites in the various

samples.

Given the large critical fluctuations, it is not surprising that the low order summations by Martinerie (1990) (see also Martinerie et al., 1990) underestimate the cut bubble correction.

The large critical fluctuations have two consequences for finite size (indeed relatively small) systems. The first is that systems of the size of actual ice samples can be simulated directly without the need to extrapolate the results to infinite size (see for example Enting, 1993). Thus simulations of the cut bubble correction should be readily achievable. The second is that the behavior of finite size systems differs from that of infinite systems by corrections termed "finite size scaling". This raises the question of whether the results shown in figure 2 could be better fitted by a finite size correction to the infinite power law behaviour (characterised by $\beta \approx 0.454$), rather than equation (1).

One of the open questions in firn closure concerns the behavior of the gas diffusivity near closure. Modelling has often assumed that this goes to zero before full closure (Trudinger et al., 2013). Percolation modeling would suggest that diffusivity behaves like conductivity of random resistor networks and go to zero at closure with a critical exponent of about 2. The extent to which the early cutoff could be reinterpreted as very slow power law decay to zero remains to be explored.

The Schaller et al paper models the system in terms of the interstices of a BCC lattice, as was done in earlier work (Enting, 1986, 1993) following a suggestion by Stauffer et al. (1985). Another open question is whether any of the universal (i.e. lattice independent ) amplitude ratios, expected in percolation transitions, can provide useful constraints on properties of firn closure, relating easily measured properties to less accessible aspects.

**References**

I. G. Enting. A lattice statistics model for the age distribution of air bubbles in polar ice. *Nature*, 315:654–655, 1985.

I. G. Enting. Lattice models of aspects of firn closure. I. percolation on the interstices of the BCC lattice. *J. Phys. A*, 19:2841–2854., 1986.

I. G. Enting. Statistics of firn closure: a simulation study. *J. Glaciology*, 39:133–142, 1993.

P. Martinerie. *Teneur en gaz des glaces polaires. Variations géographiques actuelles, variations au cours du dernier cycle climatique dans la region de Vostok*. PhD thesis, Université Joseph-Fourier, Grenoble, 1990.

P. Martinerie, V. Lipenkov, and D. Raynaud. Correction of air-content measurements in polar ice for the effect of cut bubbles at the surface of the sample. *Journal Of Glaciology*, 36:299–303, 1990.

J. Schwander and B. Stauffer. Age difference between polar ice and air trapped in its bubbles. *Nature*, 311:45–47, 1984.

B. Stauffer, J. Schwander, and H. Oeschger. Enclosure of air during the metamorphosis of dry firn to ice. *Ann. Glaciol.*, 6:108–112, 1985.

C. M. Trudinger, I. G. Enting, P. J. Rayner, D. M. Etheridge, C. Buizert, M. Rubino, P. B. Krummel, and T. Blunier. How well do different tracers constrain the firn diffusivity profile? *Atmos. Chem. Phys.*, 13:1485–1510, 2013.

Ian Enting
Honorary associate, The University of Melbourne

––––––––––––––––––––––––––––––

---

## Referee Comment (RC1) · Anonymous Referee #1 · 18 Aug 2017

General:

The authors present closed porosity data of firn and ice samples from three different polar sites in Greenland and Antarctica using 3D X-ray tomography. They find a 'universal' critical closed porosity where bubbles are sealed. While the technical approach seems robust and data are very interesting the interpretation and conclusions are too simplistic. The authors give the impression of being much closer to the physical reality than previous investigations and even accusing researchers in this field of producing corrupt data and misinterpreting them; however important details are not fully elaborated in this paper. The most important is how the scale-dependent porosity affects parameters like D-age and total air content (details in the specific comments below). The paper needs crucial revisions before it should be considered for publication.

Specific comments:

p. 1, l. 18: "direct record" seems not very informative. Probably direct records do not exist in the ice, but there is a large range of "indirectness". It should either be defined or reworded.

p. 2, l. 4: "direct measurement": the same as above.

p. 2, l. 12: "statistically solid dataset". This is a much undefined statement. Each record contains certainly a large number of data, but what does statistically solid mean? On the other hand universality of the critical porosity is deduced from only 3 records, which seem very marginal for such statement.

p. 2, l. 17: "The reduced coupling of proxies and surrounding conditions...will foster the development of new proxies, such as the air content as a marker of local insolation". This statement is somewhat unclear. I agree that it may help to put the interpretation of existing proxies on a more realistic basis, but to foster the development of new proxies is a very vague statement that calls for specific arguments.

p. 2, l. 27-30: This section requires some elaboration: "each data point": of what? "the remaining cut bubbles were less than 0.1%". How was this value determined? As the "sample" volume (1cm x 6 cm diam.) has a similar surface/volume ratio as a typical sample for porosity or total gas measurement this low value seems very surprising. Values in the order of 5-10% in the firn-ice transition zone would seem more realistic.

p. 3, l. 6: I suggest to replace "percentage" by "fraction" as the value is not given in percent

p. 4, l. 3: "Our estimation (Fig. 2b) proves a serious underestimation of the cut bubble effect and, in particular, confirms the existence of a critical porosity in contrast to recent assumptions of single-layer close-off occuring within a certain porosity range (Mitchell et al., 2015)". [occuring -> occurring]. I think there is a misunderstanding here. Mitchell et al. actually confirmed local density (or porosity) as a good predictor

for bubble closure. They only introduce stochastic variability of local density (porosity), which is well documented by measurements, to better describe the layering. But indeed there is a difference in the shape of the closed porosity (or total gas) vs. density function. Although various researchers have carefully corrected for cut bubbles an underestimation of this effect cannot be excluded. A smooth transition toward 100% closed pores as observed and still present after cut bubble correction contradicts your tomographic results an also simple percolation theory. This calls for further studies.

p. 4, l. 5-10: This paragraph needs clarification.

First, it is unfair to speak of corrupted datasets. All measured data have errors. Not all systematic errors may have been fully addressed, but therefore they are not corrupt.

Then it is most confusing to mention 37% critical closed porosity without presenting its context. This value simply relates the total gas data to the equivalent density (or porosity, or closed porosity) assuming virtual instant close-off. This has not much to do with the local pore close-off discussed here. Instead of suggesting "avoidance of such concepts" the authors should rather carefully discuss that beside the local pore close-off (at 100% local closed porosity) other factors affect total gas content in the ice (comparison with Martinerie data; Fig 3) and the concept of non- (or low-) diffusivity below a certain depth with a bulk porosity significantly above 0.1, which is crucial for the ice age – gas age difference.

p. 4, l. 13: "cannot resemble" -> "cannot fully reflect"

p. 5, l. 16-23: As mentioned above the local pore-close off is not the parameter that determines delta-age and delta-depth. It is rather the depth where diffusivity approaches zero. Better knowledge of the local close-off mechanisms is certainly very interesting but does not help to resolve the discrepancies in a simple way as suggested here.

---

## Author Comment (AC1) · 4 Sep 2017

– Please also find this response attached as a pdf. –

We would like to thank Anonymous Referee #1 for the willingness to act as a reviewer. Respectfully, however, we do not agree with several of the provided comments and the reviewer's general judgement of the manuscript. This response is an attempt to resolve these issues. We ask for this not to be seen as uncompromising - we are not unwilling to make changes and improvements to the manuscript, and have still incoorporated as many of the comments as possible - where we disagree we have explained our reasoning fully.

The referee comments are displayed in italics, followed by our responses in normal

font. For the cases where we did not follow the reviewer's suggestions, we discuss the reasons for our decision.

*General: The authors present closed porosity data of firn and ice samples from three different polar sites in Greenland and Antarctica using 3D X-ray tomography. They find a 'universal' critical closed porosity where bubbles are sealed. While the technical approach seems robust and data are very interesting the interpretation and conclusions are too simplistic. The authors give the impression of being much closer to the physical reality than previous investigations and even accusing researchers in this field of producing corrupt data and misinterpreting them; however important details are not fully elaborated in this paper. The most important is how the scale-dependent porosity affects parameters like D-age and total air content (details in the specific comments below). The paper needs crucial revisions before it should be considered for publication.*

General comment regarding accusations:

We want to clarify that we did not intend to "accuse" other researchers of anything. In our study, we show that there is a systematic error source that was underestimated in previous studies, yielding "misleading" results. Hence, we use the term "corrupt" in order to emphasize that the data do contain a systematic error, which significantly changes the results. There is no accusation in that as we are certain that this error was not underestimated or ignored intentionally.

General comment regarding the cut-bubble effect:

As we will further elaborate in the specific comments, we prove a massive underestimation of the cut-bubble effect. We are able to do so because we determine the microstructure a larger volume of firn (4 cm height, 8–10 cm diameter) and then use a sub-volume (1 cm height, 6 cm diameter) for our analysis. This way, we can determine the total and closed porosity either

1. similar to previous research (where cut bubbles would be counted as open pores as the air gets sucked out of them) or

2. eliminating the cut-bubble effect (by tracing cut bubbles within the larger volume and thus being able to decide whether they are open or closed).

The result can be seen in our manuscript, Figure 2b. The effect does not only change the values by factors of up to more than two, but also significantly influences the shape of the curve. A main reason for that is that (especially near close-off) closed parts of the pore network can extend over several centimeters, which is a classical percolation phenomenon. Previous studies estimated that correction factors of up to 1.1 would be sufficient to correct for this effect.

General comment regarding the introduction:

Several of the referee's comments do refer to "vague", "not very informative" or "undefined" statements in the introduction. We do think that the introduction is not the place to go too far into detail, but rather give a general overview of the related literature and provide a motivation for the presented study.

We'll address the specific comments below. As this only requires minor changes to the manuscript, we do not agree with the referee's demand for "crucial revisions".

*Specific comments: p. 1, l. 18: "direct record" seems not very informative. Probably direct records do not exist in the ice, but there is a large range of "indirectness". It should either be defined or reworded.*

In the scope of an introduction, we consider it worth mentioning (especially to readers that are not from the field) that the gas record is still influenced during the entrapment process. In addition, right after saying it is not a "direct record", we do further elaborate on what we intend to say with that. Similar terminology was used in published studies, e.g. Mitchell et al., 2015 "The air trapped in ice sheets is not a direct record of the past atmospheric history" (Introduction). (see also "General comment regarding the introduction")

*p. 2, l. 4: "direct measurement": the same as above.*

Actually the context is different - here we are talking about measurement methods for firn

microstructure and no longer the gas record. In contrast to previous researchers who relied on melting firn samples and deducing microstructural parameters from the results (= indirect method), we do conduct non-destructive 3D radioscopic imaging and thereby direct measurements of firn microstructure.

*p. 2, l. 12: "statistically solid dataset". This is a much undefined statement. Each record contains certainly a large number of data, but what does statistically solid mean?*

We conducted microstructure measurements throughout the lock-in zone for three cores, analyzing an unprecedented number of samples (each of a representative volume) per core. The results are very consistent and have been error-checked in various ways, amongst others comparison with other (independent) methods and repeat measurements. In contrast to previous methods, we are able to estimate uncertainties. In order to emphasize this in an introductory manner, we use the (admittedly vague) term "statistically solid". (see also "General comment regarding the introduction")

*On the other hand universality of the critical porosity is deduced from only 3 records, which seem very marginal for such statement.*

We used three cores that well reflect the temperature and accumulation rate range for polar ice cores. Previous literature would predict large differences in the critical porosity observed for these three sites, while we observe only marginal differences. Of course there is a certain probability that we by chance sampled anomalous sites / ice-cores, but we consider it highly unlikely to obtain the same anomalous value three times.

*p. 2, l. 17: "The reduced coupling of proxies and surrounding conditions...will foster the development of new proxies, such as the air content as a marker of local insolation". This statement is somewhat unclear. I agree that it may help to put the interpretation of existing proxies on a more realistic basis, but to foster the development of new proxies is a very vague statement that calls for specific arguments.*

As quoted, an example of a new proxy ("air content as a marker of local insolation") is provided.

In Chapter 5 (Implications), first paragraph, we do further elaborate on how our results will help understanding the air content and thus establishing a proxy and not only putting in on a more realistic basis. (see also "General comment regarding the introduction")

*p. 2, l. 27-30: This section requires some elaboration: "each data point": of what?*

Of the data sets presented in our study, see e.g. Table 1, Figure 1, Figure 2. We will add ".. each data point (as referred to e.g. in Table 1) ..." for clarity.

*"the remaining cut bubbles were less than $0.1\%$". How was this value determined? As the "sample" volume (1cm x 6 cm diam.) has a similar surface/volume ratio as a typical sample for porosity or total gas measurement this low value seems very surprising. Values in the order of $5 - 10\%$ in the firn-ice transition zone would seem more realistic.*

Context: "For each one meter core segment, we scanned a minimum number of five sections of approximately 4 cm height and the full core diameter (8–10 cm) with a focus on homogenous layers. [...] To eliminate the effect of cut bubbles at the surface of the sample (Martinerie et al., 1990), each data point corresponds to a layer of approximately 1 cm height and 6 cm diameter. Having the microstructure of the surrounding material in all directions at hand allows us to safely determine whether a pore is open or closed. For all measurements, the remaining cut bubbles were less than $0.1\%$ of the pore volume."

In other words:

1. We have the three-dimensional microstructure of a larger volume (4 cm height, 8–10 cm diameter) at hand.

2. We take a smaller subset (1 cm height, 6 cm diameter).

3. For most pores cut by the subset boundaries, we can still deduce whether they are open or closed because we know how they continue in the surrounding material.

4. There are some (= the "remaining") bubbles, that are part of the smaller subset, but even within the larger volume it cannot be decided whether they are open or closed.

5. We take the volume of these bubbles that lies within the subset and divide it by the total pore volume in the subset.

6. We obtain values smaller than $0.1\%$ for each sample.

Indeed, knowledge of the surrounding material is one of the main advantages of our method over previous approaches. It allows us to determine the effect of cut bubbles and show it was crucially underestimated. The value of $0.1\%$ refers to the volume fraction of pores for which we could not decide whether they were open or closed and not the volume fraction of cut bubbles.

In addition, the reviewer states that for the volume fraction of cut bubbles values in the order of $5 - 10\%$ would be realistic. However, we show (Figure 2b; Chapter "Discussion", second paragraph) that correction factors of up to $10\%$ as applied in previous studies seriously underestimate the cut bubble effect. This is a key result of our study. (see also "General comment regarding the cut-bubble effect")

*p. 3, l. 6: I suggest to replace "percentage" by "fraction" as the value is not given in percent*

Will be replaced.

*p. 4, l. 3: "Our estimation (Fig. 2b) proves a serious underestimation of the cut bubble effect and, in particular, confirms the existence of a critical porosity in contrast to recent assumptions of single-layer close-off occuring within a certain porosity range (Mitchell et al., 2015)". [occuring -> occurring]. I think there is a misunderstanding here. Mitchell et al. actually confirmed local density (or porosity) as a good predictor for bubble closure. They only introduce stochastic variability of local density (porosity), which is well documented by measurements, to better describe the layering. But indeed there is a difference in the shape of the closed porosity (or total gas) vs. density function. Although various researchers have carefully corrected for cut bubbles an underestimation of this effect cannot be excluded. A smooth transition toward $100\%$ closed pores as observed and still present after cut bubble correction contradicts your tomographic results an also simple percolation theory. This calls for further studies.*

[occuring -> occurring] will be corrected.

**CPD**
[Figure]

"Although various researchers have carefully corrected for cut bubbles an underestimation of this effect cannot be excluded." - See next-to-last answer. (see also "General comment regarding the cut-bubble effect")

Regarding Mitchell et al., 2015: We do not doubt that local density (or porosity) is a "good predictor" of bubble closure, indeed it is the determining factor. In addition, there is also nothing wrong with incorporating variability of local density to represent layering. However, they model the (local) critical porosity as a random variable, which does not seem to agree with our data. The study suffers the same problem as previously mentioned here and described in our manuscript – porosities are determined indirectly by melting/vacuumization and the cut bubble effect is only corrected for by a constant factor of 7%. Thus the closed porosity versus local density data presented show a smooth transition instead of an abrupt close-off. Mitchell et al. try to represent this in their model by making the critical porosity a random variable. However, as the data are corrupt, the model does not represent the behavior of polar firn.

*p. 4, l. 5-10: This paragraph needs clarification. First, it is unfair to speak of corrupted datasets. All measured data have errors. Not all systematic errors may have been fully addressed, but therefore they are not corrupt. Then it is most confusing to mention 37% critical closed porosity without presenting its context. This value simply relates the total gas data to the equivalent density (or porosity, or closed porosity) assuming virtual instant close-off. This has not much to do with the local pore close-off discussed here. Instead of suggesting "avoidance of such concepts" the authors should rather carefully discuss that beside the local pore close-off (at 100% local closed porosity) other factors affect total gas content in the ice (comparison with Martinerie data; Fig 3) and the concept of non- (or low-) diffusivity below a certain depth with a bulk porosity significantly above 0.1, which is crucial for the ice age – gas age difference.*

The cut bubble effect introduces errors larger than 50% near close-off (see Figure 2b). Using X-ray tomography on large volumes, we had the first opportunity to measure this effect. It turned out, it was underestimated by other researchers who did not have this opportunity. (see also "General comment regarding the cut-bubble effect" and "General comment regarding accusations")

Regarding other factors influencing the total air content - this is discussed in detail in Chapter 4, second-to-last paragraph: "Even though a single layer closes off at the same critical porosity, sealed layers may have variable air contents. Above the close-off depth, we determine average coefficients of variation for the total porosity of $1.3\%$ for B53, $1.8\%$ for B49 and $2.5\%$ for RE-CAP_S2. Higher porosity variability will lead to a larger amount of shallowly trapped bubbles, thereby increasing the air content $V$ (Stauffer et al., 1985). In our case, shallow trapping is characterized by the different slopes of the lock-in curves given in Fig. 2a, leading to increased air contents of about $2\%$ for B49 and $8\%$ for RECAP_S2 in comparison with B53. In addition, the lock-in zone extends over a depth range of approximately 7 m for B53, 9 m for B49 and 15 m for RECAP_S2. Larger lock-in zones are expected to cause enhanced sealing effects (i.e. permeable layers being sealed by impermeable ones above). This further increases the air content (Stauffer et al., 1985). The effect is hard to quantify as our measurements do not yield information about the spatial extent of horizontal layers. Nonetheless it may explain the $8\%$ and $27\%$ larger air contents for B49 and RECAP_S2 (compared to B53) respectively, that $V$ measurements for deep ice cores would predict (Martinerie et al, 1992). "

*p. 4, l. 13: "cannot resemble" -> "cannot fully reflect"*

Will be adjusted.

*p. 5, l. 16-23: As mentioned above the local pore-close off is not the parameter that determines delta-age and delta-depth. It is rather the depth where diffusivity approaches zero. Better knowledge of the local close-off mechanisms is certainly very interesting but does not help to resolve the discrepancies in a simple way as suggested here*

Even though, as stated in our manuscript, critical porosity is neither the only nor the main parameter determining delta-age or delta-depth, its temperature dependence is used in the cited delta-age calculations. Furthermore, we are aware that the simple calculations conducted in our study are not how delta-age is modeled these days. It was not our intention to do a full delta-age model, but rather estimate the dimension of the influence that avoiding the temperature-dependence introduced by Martinerie et al., 1992, has.

Please also note the supplement to this comment:
https://www.clim-past-discuss.net/cp-2017-94/cp-2017-94-AC1-supplement.pdf

————————————————————

---

## Referee Comment (RC2) · C. Buizert (Referee) · 8 Sep 2017

Schaller et al. present a new, extensive and highly valuable dataset on the bubble close-off process in polar firn, obtained using x-ray computed tomography. I would like to congratulate the authors on this achievement, which must have taken considerable time and analytical effort. The authors use this data set to provide strong observational evidence that bubble closure happens at a single porosity value, independent of the climatic conditions at the site.

Detailed observations of the close-off process are the only way to make progress on this complex problem, and I am very enthusiastic about this effort. The main experimental observation, namely that sealing of layers occurs at a constant density/porosity

value independent of the site climatic conditions, is both important and well founded in percolation theory. I am thus highly supportive of publication of this work in Climate of the Past. I give several suggestions below, which are meant to improve an already good manuscript, rather than criticize it.

My main concern is that the authors could do a better job at placing their result into a wider context, and be more respectful of previous work on this topic by avoiding phrases like "corrupted data" and "misleading". The pioneering work by Jakob Schwander, Jean-Marc Barnola and Patricia Martinerie is still relevant 25 years later, which is testimony to its quality. Rather than being "corrupted", these data simply represent measured quantities that are complementary to the micro-CT data (rather than inferior to them). For example, the casual reader of the manuscript will come away with the impression that the often-used temperature relationship by Martinerie et al. (1992, 1994) is incorrect and should be abandoned. However, Martinerie et al. studied air content, rather than firn microstructure, and I trust those data to be correct (and not "corrupted"). To me, the more interesting question is: How is it possible that air content strongly depends on the climatic conditions at the site (as demonstrated by Martinerie et al.), while the critical close-off porosity is independent of site conditions (as demonstrated by Schaller et al.). This truly is a puzzling observation, and the answer may indeed be linked to layering and interactions between adjacent layers, as the authors hint at (which are captured in air content, but not in the presented data). Presenting previous studies in this light would do justice to the quality of that work and the researchers who made those pioneering contributions.

Also, the glacial d15N problem is addressed in several locations, but not explained well. The classic reference for this problem is Landais et al. 2006, and more recently Capron et al. 2013. The issue is most obvious in d15N, with the data suggesting a thinner glacial firn column, and the models simulating a thicker one. The consequences for Dage are not as well known, mostly because there are no absolute Delta-age constraints in Antarctica to calibrate the models to, like there are in Greenland (thermal

d15N fractionation). The d15N and Dage implications are conflated in the manuscript, and could be clarified.

Specific line comments:

Title: The phrase "universal law" seems overbearing. First, the concept of universality in physics has a specific meaning, namely that near critical transitions, dynamical systems display scaling behavior that becomes independent of the details of the system being studied. This has not been demonstrated. Second, the fitting parameters (Table 2) are surprisingly different for the three sites, reducing the suggested "universality" of the behavior. I recommend that the authors revise the title of their manuscript. An example of a revised title could be: "Critical density of gas enclosure in polar firn independent of climate", or similar.

Page 1 line 5: Consider changing "universal" to "climate-independent".

Line 7: rephrase "misleading". How about: We demonstrate why indirect measurements suggest a climatic dependence

Line 10: "This may further help resolve…."

Line 22: change "safely" to "correctly"

Line 25: This is strangely formulated. The lock-in zone is commonly defined based on diffusivity (depth where d15N enrichment stops), rather than bubble closure. Of course the two overlap in depth…

Page 2 Line1 and throughout the paper: "firn model" is too vague. Specify whether you mean "firn densification model" or "firn air transport model".

Line 5: "…firn height and an indirect constraint on Dage (Sowers et al. 1992)."

Line 6: Severinghaus et al. interprets the thermal d15N signals, rather than the gravitational ones, so not the most logical citation. Also, what are the "other dating methods" referred to on line 7? For Greenland, firn densfication models do a good job based

on empirical Dage constraints from thermal d15N, see e.g. Schwander 1997, Goujon 2003, Kindler 2014, Buizert 2014, Guillevic 2013, etc.

Line 12: What does "statistically solid" mean? I would just say: "we present an extensive data set of. . ."

Line 14: replace "misleading". See my suggestion for the abstract.

Line 26: is there a reference for the Otsu method?

Line 26: please specify that you look at the pore coordination number, correct? Normally when discussing the coordination number in firn, the coordination number of the ice grains is meant.

Page 3 Line 6: *At the* percolation threshold. . ..

Line 6: I think you mean fraction rather than percentage.

Line 10-12: about the porosity range, what is this statement based on? Give a reference, or describe how this is seen in the data

Equation (1):

* Define all symbols

* The work by Mitchell et al. 2015 shows how layering can be introduced using parameterizations based on "local" (small-scale) samples to derive bulk properties. This is very important, because in modeling, firn properties are described as a function of depth, rather than porosity. When moving from porosity to depth, layering needs to be incorporated (going from local to bulk properties, in the language of Mitchell et al). Mitchell et al. use the functional form by Schwander 1989. To make the current work more accessible to firn air modelers like me, could you please try to fit the functional form of Schwander 1989, so that we can keep using the Mitchell et al. framework, but now with improved observational constraints?
\* Again for practical modeling efforts, it would further be worth having just a single best fitting curve, rather than three separate ones. Maybe that could be provided also?

\* I understand that the authors may think the last two points are an over-simplification, but please understand that it would greatly enhance the usability of your data in practical applications, which is an important motivation for doing detailed process studies like this.

\* Do you think the extensive melting at Renland could explain why that site looks so different? Even the non-melted layers were exposed to near-melting summer temperatures in the upper firn.

Line 29: This makes no sense to me. Does "extract" here refer to the collection of the sample from closed pores (usual meaning), or removal of air from open by vacuum pumping that is then discarded?

Page 4 line 2: Note that most of the Martinerie samples are done on relatively mature ice (as opposed to lock-in samples used here). In mature ice the cut bubble correction should be smaller and relatively simple as most bubbles are spherical and unconnected.

Line 5-10: I think there is some confusion in nomenclature here, as the authors point out. Close-off is not a well-defined term, and means different things to different people (which does not mean previous authors are wrong. I also don't agree with the statement that this is due to attempts to make sense of corrupted data. It is just a different approach). The Goujon/Barnola close-off is an air-content close off, i.e. the density at which the total porosity matches the air content in mature ice. From Eq. (9) in Goujon et al. it is obvious that their definition of the close-off porosity is different from the one used by the authors. I would suggest that the authors try to clarify this by using a more refined vocabulary. They could explicitly define close-off as the point at which a thin firn layer has zero open porosity, and that their definition differs from definitions used by others such as the air-content based definition by Barnola. They could e.g. refer to

their definition as the "full close-off" as opposed to the "air content close-off".

Line 20: the relation between layers reaching close-off and the extent of the lock-in zone (as defined in the gas literature as the zone between where d15N enrichments stops and the deepest pumping depth), is an interesting one. Could you elaborate, and perhaps even give some numbers?

"Sealing" is a difficult phrase, though. While diffusion is strongly inhibited in the lock-in zone, gas flow still happens. We know this because the air content in mature ice is much lower than what would be expected if the lock-in zone were really sealed. The timescale for pressure adjustment is just much shorter than that for diffusive adjustment, which means the gases are effectively sealed from diffusing following Fick's law, but not from permeating following Darcy's law (this also gives rise to dispersive mixing, as shown by Buizert and Severinghaus 2016). Since the gases diffuse and permeate through the same pore space, the only difference must be one of time scale.

Line 31: how is air content measured? Cannot be done with micro-CT, as bubble pressure is unknown.

Page 5 Line 16: This is not always true. At Greenland sites and WAIS Divide things look good.

Line 22: this makes no sense to me. How does the Vostok d15N mismatch explain 1000-2000 years? I think the d15N and Delta-age problems are conflated here. Direct Dage constraints are problematic, as Bender concludes.

Figure 2: why not show the actual DE-08 and Summit data, rather than your approximation? I think your comparison is unfair here. Typically one would apply a cut bubble correction in the range of 5-10% to those data, at which point the closed porosity of the deepest samples would go to 100% (see e.g. Fig. 3 of Mitchell et al. 2015).

Figure 3: You're comparing apples to oranges, because these are two different definitions of the close-off. I suggest you specify in the caption that you're comparing the full,

single layer close-off (your study) to the air-content (bulk) close-off (martinerie). The difference in temperature dependence is due to some poorly understood interaction between adjacent layers, not an "attempt to interpret corrupted data".

---

## Author Comment (AC2) · 4 Oct 2017

We are grateful to both referees for the time they invested to review our manuscript and think it has been greatly improved thanks to their contributions. In addition, we want to thank Ian Enting for his interesting comment that will benefit future applications of percolation modeling for the analysis of gas enclosure in polar firn. First, we would like to provide two general comments that are related to several of the reviewers' remarks. In the following, we will discuss the specific comments of both reviewers separately. The referee comments are displayed in italics, followed by our responses in normal font. For the cases where we did not follow the reviewer's suggestions, we discuss the reasons for our decision.

[Figure]

**1   General comment regarding the cut-pore effect**

In our manuscript, we prove a massive underestimation of the effect of cut pores for the melting of firn samples under vacuum conditions. We are able to do so because we measure the microstructure of a larger volume of firn (4 cm height, 8–10 cm diameter) and then use a sub-volume (1 cm height, 6 cm diameter) for our analysis. This way, we can determine the total and closed porosity either

1. similar to previous research (where cut pores would be counted as open pores as the air is removed from them) or

2. eliminating the cut-pore effect (by tracing cut pores within the larger volume and thus being able to decide whether they are open or closed).

As can be seen in our manuscript, Fig. 2b, the effect does not only change the values by factors of up to more than two, but also significantly influences the shape of the curve. A main reason for that is that (especially near close-off) closed parts of the pore network can extend over several centimeters, which is a classical percolation phenomenon (Stauffer, 1979; reference added). Previous studies estimated that additive corrections of up to $10\%$ would be sufficient to account for this effect.

It has turned out, that we did not manage to communicate this aspect of our study in the necessary clarity. In order to resolve this, we added Fig. 4 to the manuscript (also included at the end of this document) and described our approach in more detail in the "Results" section. It now reads:

"Within the microstructure analysis for the B53 core, we also mimiced the sample properties (cylindrical shape, 5 cm diameter, 5 cm height) and the method (melting the sample under vacuum, thus counting cut closed pores as part of the open pore space) as applied for Summit, Greenland (Schwander et al., 1993). This significantly changes the shape of the closed versus total porosity curve and yields results similar to previous studies (Fig. 2b). Then, by comparing

with our original data (where cut pores are traced within a larger volume to determine whether they are open or closed), we determined the necessary correction factors for the effect of cut pores (Fig. 4)."

To terminologically distinguish it from the effect of cut (near) spherical bubbles in deep ice (e.g. Martinerie et al., 1992), we do now refer to the effect as "cut-pore effect". We also tried to improve and add more details to the respective paragraph in the discussion:

"In order to estimate the closed porosity in firn, previous studies relied on measuring the amount of air enclosed in a sample by melting it in a vacuum chamber. However, during vacuumization, air is not only removed from the open pore space, but also cut closed pores, which is of particular importance for the more extensive pore network of the firn compared to deeper ice samples. Breaking of closed, but still fragile pores might even enhance this effect (Schwander and Stauffer, 1984). Nonetheless it has been neglected or only accounted for by multiplying with correction factors of up to $10\%$ to date (first applied for firn in Appendix 2 of Martinerie et al. (1992); recently Mitchell et al. (2015)). Our estimation (Fig. 2b and Fig. 4) proves a serious underestimation of the cut-pore effect. This can be explained by a classical percolation phenomenon – near the percolation threshold, individual (clusters of) closed pores can be very large compared to single bubbles (Stauffer, 1979). Indeed, we observe extents of more than a centimeter near the critical porosity for all three cores."

**2  General comment regarding previous studies**

We want to clarify that we did not intend to accuse other researchers or be disrespectful of their work. Using X-ray tomography on large volumes, we had the first opportunity to measure the cut-pore effect. It turned out, it was seriously underestimated by other researchers who did not have this opportunity. Thus, to emphasize that data sets in previous studies contain a systematic error, we (as non-native English speakers) used the term "corrupted" in the sense

"to alter from the original or correct form or version, e.g. "The file was corrupted.""
(Merriam-Webster dictionary).

As explained in the preceding "General comment regarding the cut-pore effect", the effect does not only change the closed porosity by factors of up to more than two, but also significantly influences the shape of the closed versus total porosity curve. This has led to interpretations and modeling efforts that differ from the behavior of polar firn. However, they agreed with the existing data sets and were very important scientific contributions that greatly benefitted the understanding of gas enclosure in polar firn for decades. We had hoped to adequately adress this by describing the data sets as "misleading" in the sense

"to give a wrong impression" (Merriam-Webster dictionary).

Thereby, we did not intend to critize previous authors, but, in contrast, emphasize that their interpretations were in agreement with all available data as it was impossible to know about the large impact of the cut-pore effect. As pointed out by both referees, our terminology was not the optimal choice and we did our best to adjust it in the revised version of the manuscript. We apologize and hope that no offense was taken.

**3   Anonymous Referee #1**

*General: The authors present closed porosity data of firn and ice samples from three different polar sites in Greenland and Antarctica using 3D X-ray tomography. They find a 'universal' critical closed porosity where bubbles are sealed. While the technical approach seems robust and data are very interesting the interpretation and conclusions are too simplistic. The authors give the impression of being much closer to the physical reality than previous investigations and even accusing researchers in this field of producing corrupt data and misinterpreting them;*

[Figure]

*however important details are not fully elaborated in this paper. The most important is how the scale-dependent porosity affects parameters like D-age and total air content (details in the specific comments below). The paper needs crucial revisions before it should be considered for publication.*

We would like to thank Anonymous Referee #1 for the willingness to act as a reviewer. Respectfully, however, we do not agree with some of the provided comments and the reviewer's general judgement of the manuscript. We had hoped to resolve these issues with a first short response and apologize in case it appeared too blunt. Unfortunately we did not receive a reply from the referee. Thus, we have once again reworked the manuscript to incoorporate as many of the comments as possible. We have fully explained our reasoning where we disagree and ask for this not to be seen as uncompromising - we are not unwilling to make changes and still think that the manuscript has been greatly improved thanks to the referee's comments.

We will address the specific comments below. As this only requires minor changes to the manuscript, we do not agree with the referee's demand for "crucial revisions". Please also note the two general comments (Section 1 and 2).

*Specific comments: p. 1, l. 18: "direct record" seems not very informative. Probably direct records do not exist in the ice, but there is a large range of "indirectness". It should either be defined or reworded.*

We omitted the term "direct record" here, the respective sentence now reads: "However, as bubbles are only isolated from the atmosphere at a certain depth, the firn-ice transition (50–120 m depending on the local conditions), the enclosed air is always younger than the surrounding ice."

*p. 2, l. 4: "direct measurement": the same as above.*

We respectfully disagree. The context is different - here we are talking about measurement methods for firn microstructure and no longer the gas record. In contrast to previous studies, who relied on melting firn samples under vacuum and deducing microstructural parameters

from the results (= indirect method), we conduct non-destructive 3D radioscopic imaging and thereby direct measurements of firn microstructure.

To clarify this, we are now referring to direct measurements "of firn microstructure".

*p. 2, l. 12: "statistically solid dataset". This is a much undefined statement. Each record contains certainly a large number of data, but what does statistically solid mean?*

We conducted microstructure measurements throughout the lock-in zone for three cores, analyzing an unprecedented number of samples (each of a representative volume) per core. The results are very consistent and have been error-checked in various ways, amongst others comparison with other (independent) methods and repeat measurements. In contrast to previous methods, we are able to estimate uncertainties. In order to emphasize this in an introductory manner, we used the (admittedly vague) term "statistically solid". To be more precise, we replaced it by "extensive".

*On the other hand universality of the critical porosity is deduced from only 3 records, which seem very marginal for such statement.*

We used three cores that well reflect the temperature and accumulation ranges for polar ice cores. Previous literature would predict large differences in the critical porosity for these three sites, while we observe only marginal differences. Of course there is a certain probability that we by chance sampled anomalous sites / ice cores, but we consider it highly unlikely to obtain the same anomalous value three times, especially as each data subset leading to a single data point is very large (see previous answer).

*p. 2, l. 17: "The reduced coupling of proxies and surrounding conditions...will foster the development of new proxies, such as the air content as a marker of local insolation". This statement is somewhat unclear. I agree that it may help to put the interpretation of existing proxies on a more realistic basis, but to foster the development of new proxies is a very vague statement that calls for specific arguments.*

An example of a new proxy ("air content as a marker of local insolation") is provided. In

[Figure]

Chapter 5 (Implications), first paragraph, we do further elaborate on how our results will help understanding the air content and thus establishing a proxy and not only putting in on a more realistic basis. In order to better describe this, we replaced "foster" by "benefit".

*p. 2, l. 27-30: This section requires some elaboration: "each data point": of what?*

Of the data sets presented in our study, see e.g. Table 1, Fig. 1, Fig. 2. We added ".. each data point (as referred to e.g. in Table 1) ..." for clarity.

*"the remaining cut bubbles were less than $0.1\%$". How was this value determined? As the "sample" volume (1cm x 6 cm diam.) has a similar surface/volume ratio as a typical sample for porosity or total gas measurement this low value seems very surprising. Values in the order of $5 - 10\%$ in the firn-ice transition zone would seem more realistic.*

Former context: "For each one meter core segment, we scanned a minimum number of five sections of approximately 4 cm height and the full core diameter (8–10 cm) with a focus on homogenous layers. [...] To eliminate the effect of cut bubbles at the surface of the sample (Martinerie et al., 1990), each data point corresponds to a layer of approximately 1 cm height and 6 cm diameter. Having the microstructure of the surrounding material in all directions at hand allows us to safely determine whether a pore is open or closed. For all measurements, the remaining cut bubbles were less than $0.1\%$ of the pore volume."

In other words:

1. We have the three-dimensional microstructure of a larger volume (4 cm height, 8–10 cm diameter) at hand.

2. We take a smaller subset (1 cm height, 6 cm diameter).

3. For most pores cut by the subset boundaries, we can still deduce whether they are open or closed because we know how they continue in the surrounding material.

4. There are some (= the "remaining") bubbles, that are part of the smaller subset, but even within the larger volume it cannot be decided whether they are open or closed.

5. We take the volume of these bubbles that lies within the subset and divide it by the total pore volume in the subset.

6. We obtain values smaller than $0.1\%$ for each sample.

Indeed, knowledge of the surrounding material is one of the main advantages of our method over previous approaches. It allows us to determine the effect of cut pores and show it was seriously underestimated. The value of $0.1\%$ refers to the volume fraction of pores for which we could not decide whether they were open or closed and not the volume fraction of cut pores.

In addition, the reviewer states that for the volume fraction of cut bubbles values in the order of $5 - 10\%$ would be realistic. However, we show (Fig. 2b; Chapter "Discussion", second paragraph) that correction factors of up to $10\%$ as applied in previous studies seriously underestimate the cut-pore effect. This is a key result of our study. For the various changes made to our manuscript in order to clarify this (e.g. adding Fig. 4), please see the "General comment regarding the cut-pore effect".

*p. 3, l. 6: I suggest to replace "percentage" by "fraction" as the value is not given in percent*

Replaced.

*p. 4, l. 3: "Our estimation (Fig. 2b) proves a serious underestimation of the cut bubble effect and, in particular, confirms the existence of a critical porosity in contrast to recent assumptions of single-layer close-off occuring within a certain porosity range (Mitchell et al., 2015)". [occuring -> occurring]. I think there is a misunderstanding here. Mitchell et al. actually confirmed local density (or porosity) as a good predictor for bubble closure. They only introduce stochastic variability of local density (porosity), which is well documented by measurements, to better describe the layering. But indeed there is a difference in the shape of the closed porosity (or total gas) vs. density function. Although various researchers have carefully corrected for cut bubbles an underestimation of this effect cannot be excluded. A smooth transition toward $100\%$ closed pores as observed and still present after cut bubble correction contradicts your tomographic results an also simple percolation theory. This calls for further studies.*

[occuring -> occurring] corrected.

"Although various researchers have carefully corrected for cut bubbles an underestimation of this effect cannot be excluded." - See next-to-last answer and "General comment regarding the cut-pore effect".

Regarding Mitchell et al., 2015: We do not doubt that local density (or porosity) is a "good predictor" of bubble closure, indeed it is the determining factor. In addition, there is nothing wrong with incorporating variability of local density to represent layering. However, they model the (local) critical porosity as a random variable, which does not seem to agree with our data. The study suffers the same problem as previously mentioned here and described in our manuscript – porosities are determined indirectly by melting under vacuum and the cut-pore effect is only corrected for by a constant factor of 7%. Thus the closed porosity versus local density data presented show a smooth transition instead of an abrupt close-off. Mitchell et al. try to represent this in their model by making the critical porosity a random variable. However, as the data are influenced by the cut-pore effect, the model does not represent the behavior of polar firn.

*p. 4, l. 5-10: This paragraph needs clarification. First, it is unfair to speak of corrupted datasets. All measured data have errors. Not all systematic errors may have been fully addressed, but therefore they are not corrupt. Then it is most confusing to mention 37% critical closed porosity without presenting its context. This value simply relates the total gas data to the equivalent density (or porosity, or closed porosity) assuming virtual instant close-off. This has not much to do with the local pore close-off discussed here. Instead of suggesting "avoidance of such concepts" the authors should rather carefully discuss that beside the local pore close-off (at 100% local closed porosity) other factors affect total gas content in the ice (comparison with Martinerie data; Fig 3) and the concept of non- (or low-) diffusivity below a certain depth with a bulk porosity significantly above 0.1, which is crucial for the ice age – gas age difference.*

The cut-pore effect influences closed porosity by more than a factor of two near close-off (see Fig. 2b and the new Fig. 4). Using X-ray tomography on large volumes, we had the first opportunity to measure this effect. It turned out, it was underestimated by other researchers who

did not have this opportunity. (Please see also "General comment regarding the cut-pore effect" and "General comment regarding previous studies".)

Regarding other factors influencing the total air content - this was discussed in detail in Chapter 4, second-to-last paragraph, where we added further details:

"Even though a single layer closes off at the same critical porosity, sealed layers may have variable air contents. Above the close-off depth, we determine average coefficients of variation for the total porosity of $1.3\%$ for B53, $1.8\%$ for B49 and $2.5\%$ for RECAP_S2. Higher porosity variability will lead to a larger amount of shallowly trapped bubbles, thereby increasing the air content $V$ (Stauffer et al., 1985). In our case, the effect of shallow trapping can be estimated from the different slopes of the lock-in curves given in Fig. 2a, yielding possible increases in air content of about $2\%$ for B49 and $8\%$ for RECAP_S2 in comparison with B53. This implicitly assumes that closed and open porosity undergo the same compaction as the firn densifies and thus has to be interpreted as the maximum possible influence of shallow trapping. In addition, the lock-in zone extends over a depth range of approximately 7 m for B53, 9 m for B49 and 15 m for RECAP_S2. Larger lock-in zones are expected to cause enhanced sealing effects (i.e. permeable layers being sealed by impermeable ones above). This further increases the air content (Stauffer et al., 1985). The effect is hard to quantify as our measurements do not yield information about the spatial extent of horizontal layers and it does not take into account pressure adjustment within the lock-in zone which is happening on a much shorter time scale compared to diffusion (Buizert and Severinghaus, 2016). Nonetheless, it may explain the $8\%$ and $27\%$ larger air contents for B49 and RECAP_S2 (compared to B53) respectively, that $V$ measurements for deep ice cores would predict according to the observed temperature dependence (Martinerie et al, 1992). In return, even though we do not observe this temperature dependence for the gas enclosure within single layers, it is a signal that seems to originate from the lock-in zone, presumably as a consequence of a distinct density layering."

To account for this, the quoted paragraph now directly follows the discussion of previous approaches (such as the $37\%$ critical porosity).
Finally, the respective paragraph has been rewritten and now reads:

"While concepts such as gas enclosure (both in single layers and as a bulk property) occurring at a critical closed (Goujon et al., 2003) or open (Gregory et al., 2014) porosity have become widely accepted, they do not seem to agree with the results of our firn microstructure analysis and the previously discussed (conceptual) definitions of the lock-in zone. As a consequence, refinement of these theories may greatly benefit the understanding of gas enclosure in polar firn. Notably, the critical closed porosity value of 37% identified by Jean-Marc Barnola using porosity measurements of several ice cores from Greenland and Antarctica (Goujon et al., 2003) corresponds to a total porosity of approximately 0.1 for the two data sets (Summit and B53) that are affected by the cut-pore effect (displayed in Fig. 2b)."

*p. 4, l. 13: "cannot resemble" -> "cannot fully reflect"*

Adjusted.

*p. 5, l. 16-23: As mentioned above the local pore-close off is not the parameter that determines delta-age and delta-depth. It is rather the depth where diffusivity approaches zero. Better knowledge of the local close-off mechanisms is certainly very interesting but does not help to resolve the discrepancies in a simple way as suggested here*

Even though, as stated in our manuscript, critical porosity is neither the only nor the main parameter determining $\Delta$age or $\Delta$depth, its temperature dependence is used in the cited $\Delta$age calculations. Furthermore, we are aware that the simple calculations conducted in our study are not how $\Delta$age is modeled these days. It was not our intention to do a full $\Delta$age model, but rather estimate the dimension of the influence that avoiding the temperature-dependence introduced by Martinerie et al. (1992) has. In order to better represent this, we now also state we "estimate" (instead of "calculate") the change in $\Delta$age.

**4 Christo Buizert**

*Schaller et al. present a new, extensive and highly valuable dataset on the bubble close-off process in polar firn, obtained using x-ray computed tomography. I would like to congratulate the authors on this achievement, which must have taken considerable time and analytical effort. The authors use this data set to provide strong observational evidence that bubble closure happens at a single porosity value, independent of the climatic conditions at the site.*

*Detailed observations of the close-off process are the only way to make progress on this complex problem, and I am very enthusiastic about this effort. The main experimental observation, namely that sealing of layers occurs at a constant density/porosity value independent of the site climatic conditions, is both important and well founded in percolation theory. I am thus highly supportive of publication of this work in Climate of the Past. I give several suggestions below, which are meant to improve an already good manuscript, rather than criticize it.*

We would like to thank the reviewer, Christo Buizert, for his kind words and very helpful comments. We have responded to all of them below and tried to address as many as possible to significantly improve the manuscript. The referee comments are displayed in italics, followed by our responses in normal font. For the few cases where we did not follow the reviewer's suggestions, we discuss the reasons for our decision.

*My main concern is that the authors could do a better job at placing their result into a wider context, and be more respectful of previous work on this topic by avoiding phrases like "corrupted data" and "misleading". The pioneering work by Jakob Schwander, Jean-Marc Barnola and Patricia Martinerie is still relevant 25 years later, which is testimony to its quality. Rather than being "corrupted", these data simply represent measured quantities that are complementary to the micro-CT data (rather than inferior to them). For example, the casual reader of the manuscript will come away with the impression that the often-used temperature relationship by Martinerie et al. (1992, 1994) is incorrect and should be abandoned. However, Martinerie et al. studied air content, rather than firn microstructure, and I trust those data to be correct (and not "corrupted"). To me, the more interesting question is: How is it possible that air*

*content strongly depends on the climatic conditions at the site (as demonstrated by Martinerie et al.), while the critical close-off porosity is independent of site conditions (as demonstrated by Schaller et al.). This truly is a puzzling observation, and the answer may indeed be linked to layering and interactions between adjacent layers, as the authors hint at (which are captured in air content, but not in the presented data). Presenting previous studies in this light would do justice to the quality of that work and the researchers who made those pioneering contributions.*

We have reworked the presentation of previous studies in order to do justice to their quality. For details, see "General comment regarding previous studies".

*Also, the glacial d15N problem is addressed in several locations, but not explained well. The classic reference for this problem is Landais et al. 2006, and more recently Capron et al. 2013. The issue is most obvious in d15N, with the data suggesting a thinner glacial firn column, and the models simulating a thicker one. The consequences for Dage are not as well known, mostly because there are no absolute Delta-age constraints in Antarctica to calibrate the models to, like there are in Greenland (thermal d15N fractionation). The d15N and Dage implications are conflated in the manuscript, and could be clarified.*

We have included the mentioned references (in both introduction/discussion) and reworked the respective introductory paragraph. It now reads:

"The $\delta^{15}N$ of $N_2$ has been established as a proxy for firn height and thus an indirect constraint on $\Delta$age (Sowers et al., 1992). This relation has successfully been tested for high-accumulation sites, e.g. the last 40,000 years at Summit, Greenland (Schwander et al., 1997). On the contrary, there is a mismatch of up to 2,000 years with model results for the East Antarctic plateau (Bender et al., 2006; Parrenin et al., 2012). These modeled chronologies are based on the current knowledge of bubble trapping in polar firn and particularly sensitive to the critical porosity via the assumed temperature dependence. Deviations from the simple relationships used to reconstruct past temperatures and accumulation rates from the water isotopic composition have been suggested as a possible explanation (Landais et al., 2006), while the hypothesis of a large glacial convective zone as an important factor has been ruled out (Capron et al., 2013). Recently the inclusion of impurity effects has reduced the mismatch for East Antarctic sites, however it deteriorates the agreement between modelled and measured $\delta^{15}$N for high-accumulation sites (Breant et al., 2017)."

*Specific line comments: Title: The phrase "universal law" seems overbearing. First, the concept of universality in physics has a specific meaning, namely that near critical transitions, dynamical systems display scaling behavior that becomes independent of the details of the system being studied. This has not been demonstrated. Second, the fitting parameters (Table 2) are surprisingly different for the three sites, reducing the suggested "universality" of the behavior. I recommend that the authors revise the title of their manuscript. An example of a revised title could be: "Critical density of gas enclosure in polar firn independent of climate", or similar.*

Adjusted.

*Page 1 line 5: Consider changing "universal" to "climate-independent"*

Adjusted.

*Line 7: rephrase "misleading". How about: We demonstrate why indirect measurements suggest a climatic dependence*

Adjusted.

*Line 10: "This may further help resolve..."*

Adjusted.

*Line 22: change "safely" to "correctly"*

Adjusted.

*Line 25: This is strangely formulated. The lock-in zone is commonly defined based on diffusivity (depth where d15N enrichment stops), rather than bubble closure. Of course the two overlap in depth...*

We respectfully disagree with that being "commonly defined". For example, Bender et. al

(2006) state "Below the diffusive zone lies the lock-in zone. Here, alternating layers of open and closed ice preserve some open porosity. Concentrations (or isotopic compositions) remain constant within individual layers, but open porosity allows firn air sampling. The depth where all porosity is closed corresponds to the bottom of the lock-in zone."

However, we think the (seemingly different) definitions do actually coincide and have reworked and extended the following paragraph to clarify the terminology:

"The problem of understanding gas enclosure in polar firn has been tackled with different methods and from various perspectives such as firn microstructure, firn air transport and firn air pumpings. As a consequence, seemingly different definitions have been established for terminological frameworks such as the "lock-in zone". The results of two firn air pumpings conducted at the RECAP drill site (T. Sowers, personal communication, 2017) and at Kohnen station, close to B49 (Weiler, 2008) in combination with high-resolution X-ray porosity measurements (Freitag et al., 2013), corroborate our microstructural findings. For both sites, the sharp decline in $CO_2$, $CH_4$ and $N_2O$ concentrations (interpreted as the onset of the lock-in zone according to firn air pumpings) coincides with the occurrence of the first significant (i.e. at least 1 cm thick) layer with a porosity below the critical value of $0.1$. On the other hand, no more air can be pumped (bottom of the lock-in zone according to firn air pumpings) when there are no further layers with a total porosity larger than $0.1$. In the firn air transport literature (e.g. Buizert and Severinghaus, 2016), the onset of the lock-in zone (also refered to as "lock-in depth") is defined as the depth where molecular diffusion effectively ceases. According to percolation theory this happens at the percolation threshold, i.e. the point when there is no longer a connected component of the order of the system size (Ghanbarian and Hunt, 2014). This corresponds to the first layer reaching a closed porosity of $100\%$, which is the onset of the lock-in zone in the microstructural sense. Regarding the bottom of the lock-in zone, it has been observed that due to vertical mixing the air composition in a certain depth does only no longer change (definition according to gas transport) at the "close-off depth" (Buizert et al., 2012). It is defined as the depth at which all pores are closed (Witrant et al., 2012) and thereby also coincides with the bottom of the lock-in zone according to firn microstructure. Thus, the three definitions for the lock-in

zone (according to firn microstructure, firn air transport and firn air pumpings) are equivalent. Furthermore, the limits of the lock-in zone are solely determined by the existence of significant layers above and below the critical porosity, and thereby the (cm-scale) porosity variability."

*Page 2 Line1 and throughout the paper: "firn model" is too vague. Specify whether you mean "firn densification model" or "firn air transport model"*

All four occurrences have been changed to "firn air model", a term already used in previous literature (e.g. Mitchell et al., 2015) to refer to both air transport and age dating.

*Line 5: "...firn height and an indirect constraint on Dage (Sowers et al. 1992)."*

Changed to "... firn height and thus an indirect constraint on ..."

*Line 6: Severinghaus et al. interprets the thermal d15N signals, rather than the gravitational ones, so not the most logical citation. Also, what are the "other dating methods" referred to on line 7? For Greenland, firn densfication models do a good job based on empirical Dage constraints from thermal d15N, see e.g. Schwander 1997, Goujon 2003, Kindler 2014, Buizert 2014, Guillevic 2013, etc.*

We agree, thus changed to: "This relation has successfully been tested for high-accumulation sites, e.g. the last 40,000 years at Summit, Greenland (Schwander et al., 1997)."

Please note: Schwander et al. (1997) do interpret the gravitational $\delta^{15}N$ signal. Samples that reflect thermal fractionation due to rapid warming were excluded from the study.

*Line 12: What does "statistically solid" mean? I would just say: "we present an extensive data set of..."*

Adjusted here and in the abstract.

*Line 14: replace "misleading". See my suggestion for the abstract.*

Replaced by "different".

*Line 26: is there a reference for the Otsu method?*

"Otsu (1979)" has been added as a reference.

*Line 26: please specify that you look at the pore coordination number, correct? Normally when discussing the coordination number in firn, the coordination number of the ice grains is meant.*

Adjusted.

*Page 3 Line 6: \*At the\* percolation threshold...*

The reviewer's suggestion would change the meaning of the sentence. For clarification, we rephrased it to:

"For this lattice, the fraction of channels occupied by air at close-off, the so-called percolation threshold, is known to be ..."

*Line 6: I think you mean fraction rather than percentage.*

Yes, corrected.

*Line 10-12: about the porosity range, what is this statement based on? Give a reference, or describe how this is seen in the data*

As stated, it can be seen in the much steeper slope of the closed porosity curve in Fig. 2a. However, we agree that the corresponding sentence was not well-structured logically and thus rephrased it to:

"However, as indicated by the much steeper slope of the closed porosity, enclosure takes place in a significantly smaller porosity range for the East Antarctic cores compared to the coastal Greenland site."

*Equation (1): \* Define all symbols*

Done.

*\* The work by Mitchell et al. 2015 shows how layering can be introduced using parameterizations based on "local" (small-scale) samples to derive bulk properties. This is very important, because in modeling, firn properties are described as a function of depth, rather than poros-*

*ity. When moving from porosity to depth, layering needs to be incorporated (going from local to bulk properties, in the language of Mitchell et al). Mitchell et al. use the functional form by Schwander 1989. To make the current work more accessible to firn air modelers like me, could you please try to fit the functional form of Schwander 1989, so that we can keep using the Mitchell et al. framework, but now with improved observational constraints?*

In general, we think that it would be premature to go ahead and use our current results ("only" three sites after all) to parametrize any functional form based on climatic conditions, e.g. temperature or accumulation rate for the exponential decay factors ($\lambda$ in the Schwander parametrization, $\lambda_1$ and $\lambda_2$ here). Nonetheless, we plan to further investigate possible parametrizations of a functional form as we analyze gas enclosure for more polar ice cores (in particular a site around -30C – probably a firn core from the East Greenland Ice-core Project). Specifically, our results are available through PANGAEA, such that individual modelers can fit any functional form depending on their needs.

*\* Again for practical modeling efforts, it would further be worth having just a single best fitting curve, rather than three separate ones. Maybe that could be provided also?*

*\* I understand that the authors may think the last two points are an over-simplification, but please understand that it would greatly enhance the usability of your data in practical applications, which is an important motivation for doing detailed process studies like this.*

This was done, see Fig. 2a - "Schwander model ($\Phi_{\text{crit}} = 0.1$)" and Discussion, fourth paragraph:

"Remarkably, for the correct critical porosity [i.e. 0.1], the Schwander parametrization (Schwander, 1989) seems to approximately represent a site-independent average relation of closed and total porosity (cf. Fig. 2a). However, due to the lack of other parameters, it cannot resemble the behavior of polar firn. Therefore we decided to derive a more complex exponential-decay relation (Eq. (1)) to fit our results."

Here, we included an additional reference to Fig. 2a.

*\* Do you think the extensive melting at Renland could explain why that site looks so different?*

*Even the non-melted layers were exposed to near-melting summer temperatures in the upper firn.*

Apart from the Renland data, the differences between B49 and B53 may also indicate a dependence of the slope on the climatic conditions. However, based on the existing data, one can only speculate about such aspects. It will be necessary to analyze both further firn cores throughout the temperature (and accumulation) range and the microstructural properties and layering of the snowpack at the drilling sites.

*Line 29: This makes no sense to me. Does "extract" here refer to the collection of the sample from closed pores (usual meaning), or removal of air from open by vacuum pumping that is then discarded?*

The latter. In order to clarify we rephrased the corresponding sentences to: "In order to estimate the closed porosity in firn, previous studies relied on measuring the amount of air enclosed in a sample by melting it in a vacuum chamber. However, during vacuumization, air is not only removed from the open pore space, but also cut closed pores, which is of particular importance for the more extensive pore network of the firn compared to deeper ice samples. Breaking of closed, but still fragile pores might even enhance this effect..."

*Page 4 line 2: Note that most of the Martinerie samples are done on relatively mature ice (as opposed to lock-in samples used here). In mature ice the cut bubble correction should be smaller and relatively simple as most bubbles are spherical and unconnected.*

Please note that Martinerie et al. (1992) also compare their results to closed pore volume measurements of firn samples from Summit and Siple (Chapter 4.3). and as they discuss in Appendix 2, apply a correction factor of $10\%$ for the effect of cut pores in firn firn samples. To better account for this, we detailed the respective reference and clarified in the surrounding paragraph that we are only discussing the cut-pore effect in firn (and not deep ice). It now reads:

"In order to estimate the closed porosity in firn, previous studies relied on measuring the amount

of air enclosed in a sample by melting it in a vacuum chamber. However, during vacuumization, air is not only removed from the open pore space, but also cut closed pores, which is of particular importance for the more extensive pore network of the firn compared to deeper ice samples. Breaking of closed, but still fragile pores might even enhance this effect (Schwander and Stauffer, 1984). Nonetheless it has been neglected or only accounted for by multiplying with correction factors of up to $10\%$ to date (first applied for firn in Appendix 2 of Martinerie et al. (1992); recently Mitchell et al. (2015))."

We think that, regarding the methodology, our references to Martinerie et al. (1992) are adequate, see e.g. Discussion, first paragraph: "In previous literature, ice densities at air isolation level were obtained from air-content measurements on deep ice samples (Martinerie et al., 1992).", but added another reference in order to clarify that we do not doubt their main result, the temperature dependence of $V$:

"...that $V$ measurements for deep ice cores would predict according to the observed temperature dependence (Martinerie et al., 1992). In return, even though we do not observe this temperature dependence for the gas enclosure within single layers, it is a signal that seems to originate from the lock-in zone, presumably as a consequence of a distinct density layering."

*Line 5-10: I think there is some confusion in nomenclature here, as the authors point out. Close-off is not a well-defined term, and means different things to different people (which does not mean previous authors are wrong. I also don't agree with the statement that this is due to attempts to make sense of corrupted data. It is just a different approach). The Goujon/Barnola close-off is an air-content close off, i.e. the density at which the total porosity matches the air content in mature ice. From Eq. (9) in Goujon et al. it is obvious that their definition of the close-off porosity is different from the one used by the authors. I would suggest that the authors try to clarify this by using a more refined vocabulary. They could explicitly define close-off as the point at which a thin firn layer has zero open porosity, and that their definition differs from definitions used by others such as the air-content based definition by Barnola. They could e.g. refer to their definition as the "full close-off" as opposed to the "air content close-off".*

We avoided usage of the term "close-off" – instead we are now referring to "gas enclosure in/within a single layer". In addition, we have reworked our terminology with respect to previous approaches (see also "General comment regarding previous studies"). The respective paragraph has been split up and rephrased, the part that refers to Goujon et al. (2003) now reads:

"While concepts such as gas enclosure (both in single layers and as a bulk property) occurring at a critical closed (Goujon et al., 2003) or open (Gregory et al., 2014) porosity have become widely accepted, they do not seem to agree with the results of our firn microstructure analysis and the previously discussed (conceptual) definitions of the lock-in zone. As a consequence, refinement of these theories may greatly benefit the understanding of gas enclosure in polar firn. Notably, the critical closed porosity value of 37% identified by Jean-Marc Barnola using porosity measurements of several ice cores from Greenland and Antarctica (Goujon et al., 2003) corresponds to a total porosity of approximately 0.1 for the two data sets (Summit and B53) that are affected by the cut-pore effect (displayed in Fig. 2b)."

We are not stating that Goujon et al. (2003) do apply the same definitions as we do, we just observe that, interestingly, for both cut-pore-affected data sets (Fig. 2b) a closed porosity of 37% corresponds to a total porosity of 0.1 (i.e. the critical porosity we identified for the gas enclosure in a single layer). Furthermore, they state (in the paragraph following Eq. (9)), that a closed porosity of 37% would correspond to the bottom of the non-diffusive zone (and thereby not the lock-in depth but the close-off depth). Nonetheless, in order to be consistent with the previously discussed definitions of the lock-in zone (and, in particular, the close-off depth) this has to be 100% both locally and as a bulk property.

*Line 20: the relation between layers reaching close-off and the extent of the lock-in zone (as defined in the gas literature as the zone between where d15N enrichments stops and the deepest pumping depth), is an interesting one. Could you elaborate, and perhaps even give some numbers?*

Unfortunately, we do not have any $\delta^{15}$N of $N_2$ data. However, (as previously discussed here and

added to the manuscript) the definitions of the lock-in zone according to microstructure and firn air transport do actually coincide. According to the microstructural definition, further details (and numbers) are provided in the following paragraph, e.g. "... the lock-in zone extends over a depth range of approximately 7 m for B53, 9 m for B49 and 15 m for RECAP_S2.".

*"Sealing" is a difficult phrase, though. While diffusion is strongly inhibited in the lock-in zone, gas flow still happens. We know this because the air content in mature ice is much lower than what would be expected if the lock-in zone were really sealed. The timescale for pressure adjustment is just much shorter than that for diffusive adjustment, which means the gases are effectively sealed from diffusing following Fick's law, but not from permeating following Darcy's law (this also gives rise to dispersive mixing, as shown by Buizert and Severinghaus 2016). Since the gases diffuse and permeate through the same pore space, the only difference must be one of time scale.*

True, we included this: "However, the effect is hard to quantify as our measurements do not yield information about the spatial extent of horizontal layers and it does not take into account pressure adjustment within the lock-in zone which is happening on a much shorter time scale compared to diffusion (Buizert and Severinghaus, 2016)."

*Line 31: how is air content measured? Cannot be done with micro-CT, as bubble pressure is unknown.*

Air content was not measured. Only the effect of shallow trapping on the air content was estimated from the different slopes of the closed versus total porosity curves, which implicitly assumes that closed and open porosity undergo the same compaction as the firn densifies. To clarify this, we rephrased the corresponding sentence to: "In our case, the effect of shallow trapping can be estimated from the different slopes of the lock-in curves given in Fig. 2a, yielding possible increases in air content of about $2\%$ for B49 and $8\%$ for RECAP_S2 in comparison with B53. This implicitly assumes that closed and open porosity undergo the same compaction as the firn densifies and thus has to be interpreted as the maximum possible influence of shallow trapping."

*Page 5 Line 16: This is not always true. At Greenland sites and WAIS Divide things look good.*

Agreed, we added "for the East Antarctic plateau" for clarity.

*Line 22: this makes no sense to me. How does the Vostok d15N mismatch explain 1000-2000 years? I think the d15N and Delta-age problems are conflated here. Direct Dage constraints are problematic, as Bender concludes.*

The other way around: Using a constant critical porosity of 0.1 instead of the Martinerie et al. (1992) relation for our simple $\Delta$age estimations, we reduce the mismatch with $\delta^{15}$N of $N_2$ by more than 1,000 years. To clarify this, we changed the corresponding sentence from

"On average, this reduces the gas age–ice age difference ..." to

"On average, excluding the temperature dependence of the critical porosity reduces the gas age–ice age difference by well over $10\%$.".

*Figure 2: why not show the actual DE-08 and Summit data, rather than your approximation? I think your comparison is unfair here. Typically one would apply a cut bubble correction in the range of $5-10\%$ to those data, at which point the closed porosity of the deepest samples would go to $100\%$ (see e.g. Fig. 3 of Mitchell et al. 2015).*

We included the actual Summit data (similar shape as the DE-08 data) to better compare with the B53 data (different shape). Application of a correction in the range of $5-10\%$ leaves the shape of the curve unchanged. In contrast, one would need much larger correction factors (see "General comment regarding the cut-pore effect" and the newly added Fig. 4).

*Figure 3: You're comparing apples to oranges, because these are two different definitions of the close-off. I suggest you specify in the caption that you're comparing the full, single layer close-off (your study) to the air-content (bulk) close-off (martinerie). The difference in temperature dependence is due to some poorly understood interaction between adjacent layers, not an "attempt to interpret corrupted data".*

We have added more detail to the caption, it now reads

"Critical porosity versus temperature. The given linear relation is commonly fit to the data of Martinerie et al. (1992), a study based on air-content measurements of 495 deep ice samples from sixteen cores (with a minimum of only two measurements for one core). From their results, they reconstruct the average ice density at air isolation which is equivalent to the critical porosities shown here. In contrast, we analyzed the microstructure of 1163 firn samples for three cores (see Table 1), allowing the direct determination of the critical porosity of gas enclosure within a single layer.".

Furthermore, we made several (previously discussed) changes when referring to Martinerie et al. (1992) and after the reference to Fig. 3 within the text, we added

"For the gas enclosure within single layers, we do not observe the commonly assumed temperature dependence ...".

**Changes in manuscript**

Please find an updated version of the manuscript (changes tracked with latexdiff) attached.

Please also note the supplement to this comment:
https://www.clim-past-discuss.net/cp-2017-94/cp-2017-94-AC2-supplement.pdf

————————————————

[Figure]

**Fig. 1.**

**Supplement:**

[revised manuscript text omitted]